# PMI-controlled mannose metabolism and glycosylation determines tissue tolerance and virus fitness

Ronghui Liang[1,2,10], Zi-Wei Ye[3,10], Zhenzhi Qin[2,10], Yubin Xie[2], Xiaomeng Yang[2], Haoran Sun[2,4], Qiaohui Du[5], Peng Luo [2], Kaiming Tang[2], Bodan Hu[2,6], Jianli Cao[2], Xavier Hoi-Leong Wong [7], Guang-Sheng Ling [3], Hin Chu [2,4,6,8], Jiangang Shen[5], Feifei Yin[1], Dong-Yan Jin [3,6,8], Jasper Fuk-Woo Chan [1,2,4,6,8,9], Kwok-Yung Yuen [1,2,4,6,8,9,11] & Shuofeng Yuan [2,4,6,8,11] ✉

Host survival depends on the elimination of virus and mitigation of tissue damage. Herein, we report the modulation of D-mannose flux rewires the virus-triggered immunometabolic response cascade and reduces tissue damage. Safe and inexpensive D-mannose can compete with glucose for the same transporter and hexokinase. Such competitions suppress glycolysis, reduce mitochondrial reactive-oxygen-species and succinate-mediated hypoxia-inducible factor-1α, and thus reduce virus-induced proinflammatory cytokine production. The combinatorial treatment by D-mannose and antiviral monotherapy exhibits in vivo synergy despite delayed antiviral treatment in mouse model of virus infections. Phosphomannose isomerase (*PMI*) knockout cells are viable, whereas addition of D-mannose to the *PMI* knockout cells blocks cell proliferation, indicating that PMI activity determines the beneficial effect of D-mannose. PMI inhibition suppress a panel of virus replication via affecting host and viral surface protein glycosylation. However, D-mannose does not suppress PMI activity or virus fitness. Taken together, PMI-centered therapeutic strategy clears virus infection while D-mannose treatment reprograms glycolysis for control of collateral damage.

The twenty-first century has already recorded more than ten emerging viral epidemics or pandemic including the COVID-19. As the emergence of such novel epidemic virus is expected to accelerate, there is an increasing need for strategic approaches to develop broadly active and multimodal therapeutics ready for immediate use in future epidemics. Identifying combination therapies that can target both viruses and immune-mediated pathologies, prolong the therapeutic window after the onset of symptoms, and lower the risk of drug resistance is crucial. Defense against viral infections in hosts involves mechanisms of both resistance and tolerance. Resistance decreases the viral burden once an infection has taken hold, while tolerance lessens the adverse effects of an infection on the host's well-being without directly influencing the viral load. Disease severity can stem from compromised host resistance, which is marked by a high viral load, or from inadequate host tolerance. Immunopathology belongs to the latter group, and inadequate tissue protection is likely a significant factor in conditions marked by excessive inflammation, like sepsis[1,2]. Research on immunometabolism has shown that both intracellular and systemic metabolic processes affect immune cells. Alterations in a host's nutritional status can impact effector cell functions and the outcome of infections. This is because immune responses against virus infections require a significant amount of energy, involve metabolic shifts, and demand specific nutrients. Indeed, remodeling of lipid metabolism renders protection against virus infection in vitro and in vivo[3,4].

Glucose plays crucial roles in cellular energy production, storage, and regulation. It is also involved in the development of pathological conditions after virus infection. For instance, when human monocytes infected with SARS-CoV-2 are cultured under high glucose conditions, their metabolism is rewired. This significantly increases viral replication and cytokine production, which in turn weakens the T-cell response and impairs T-cell function[5]. Inhibition of glycolysis using 2-deoxy-D-glucose (2-DG) or by inhibiting glycolytic enzymes such as 6-phospho-fructo-2-kinase/fructose-2,6-bisphosphatase-3 (PFKFB3) eliminates replication of a panel of different viruses, which suggests that glycolysis is a crucial upstream event in multiple virus pathogenesis[6]. Suppression of glycolytic enzymes, such as glyceraldehyde-3-phosphate dehydrogenase and aldolase A, significantly contributes to the anti-inflammatory effects produced by the immunosuppressive metabolite itaconate[7] and fumarate[8] in macrophages. Overall, these findings strongly highlight the therapeutic potential of modulating glucose metabolism as a promising approach for controlling infectious and inflammatory diseases.

D-mannose (hereafter referred to as mannose), a C-2 epimer of glucose, has a physiological blood concentration less than one-fiftieth of that of glucose. Mannose effectively blocks bacterial adhesion by binding to these pili, which stops bacteria from colonizing the urinary tract and causing infections[9]. Recently, mannose is reported to counteract lipopolysaccharide (LPS)-induced macrophage activation by negatively impacting interleukin 1β (Il1b) gene expression[10]. This monosaccharide can either be converted into glucose for catabolism or be derived from glucose for glycan biosynthesis. Mannose is transported into mammalian cells through facilitated diffusion by glucose transporters (GLUT). Once inside the cell, mannose is phosphorylated by hexokinase to produce mannose-6-phosphate (M6P), which can follow two primary metabolic pathways. A small portion of M6P is converted to mannose-1-phosphate by phosphomannomutase, which then enters glycosylation pathways. The majority of M6P is converted to fructose-6-phosphate (F6P) by phosphomannose isomerase (PMI). M6P has the ability to inhibit three enzymes involved in glucose metabolism: hexokinases, phosphoglucose isomerase (PGI), and glucose-6-phosphate dehydrogenase[11]. This interaction highlights the interconnected nature of sugar metabolism pathways in cells.

Here, we affirmed that glycolytic flux is required for the replication of a broad range of medically important but phylogenetically distinct viruses, including SARS-CoV-2, influenza A virus (IAV) H1N1 and zika virus (ZIKV). Using SARS-CoV-2 and H1N1 as models, we found that the H1N1-induced mitochondrial reactive oxygen species (mtROS) production as well as HIF-1α activation, which in turn upregulates IL-1β expression, whereas mannose treatment rewires such ROS-induced mitochondrial dysfunction via modulation of glycolysis and tricarboxylic acid (TCA) cycle. These findings may help explain why uncontrolled diabetes impairs the adaptive immune response and contributes to lung dysfunction in patients with severe influenza or COVID-19 symptoms[12]. The data also provide mechanistic evidence that suggests targeting the glycolysis/mtROS/HIF-1α axis could be a potential modulatory approach to improve tissue tolerance following respiratory virus infections. Moreover, we pinpointed the versatility of PMI not only as a gatekeeper enzyme dictating the mannose metabolism, but also a druggable target that affects virus/host protein glycosylation and therefore virus entry. Our study paves a way to modulate host sugar metabolism for elevated host tolerance of inflammatory damage and a potential strategy to target PMI-mediated glycosylation for the development of next-generation antiviral therapy.

## Results

### Inhibition of glucose utilization by mannose is protective

We have previously demonstrated the importance of lipogenesis pathways during virus life cycles[3,4]. Following this strategy, we used influenza A H1N1 virus (pdm09) as the model for proof of concept. We performed targeted quantitation of polar metabolites on the human bronchial and tracheal epithelial cells (hBTEC) (Supplementary Fig. 1). Similar findings have been reported on human neural progenitor cells (hNPC) infected by enterovirus A71[13]. These results demonstrated that host glucose metabolism is commonly altered upon both virus infections. The glycolytic pathway begins with the glucose or mannose which is converted into pyruvate and eventually enters the tricarboxylic acid cycle. As a competitor of glucose for hexokinase (HK) enzyme, mannose enters glycolysis as well, through the common GLUT family or mannose-specific transporter[14,15], first being converted to mannose-6-phosphate and then isomerized to produce fructose-6-phosphate by Mannose-6-phosphate isomerase (PMI). Of note, glycolysis suppression by a pharmacological inhibitor 2-DG has been shown to decrease the replication of a panel of viruses including SARS-CoV-2 and influenza A virus[16,17]. To determine if mannose renders protection against virus infection, we first demonstrated that mannose treatment in mice is safe, whether via intraperitoneal (I.P.), intravenous (I.V.), or oral (P.O.) administration (Supplementary Fig. 2a). We then employed three lethal mouse models for evaluation of survival rates after virus infection and mannose treatment (Fig. 1a). Mice were orally administrated with 2 g/kg/day of mannose after lethal challenge with each virus (five lethal dose 50%) until 6 days post-infection (dpi), followed by health surveillance until 14dpi. In a K18-hACE2 mouse model of SARS-CoV-2, mice receiving mannose therapy showed better survival rate (20% vs 0%) and extended median survival date (8 vs 6) than that of PBS control group (Fig. 1b). Protection against lethal challenge with influenza A H1N1 in BALB/c mice was also achievable, which is evidenced by the significantly increased survival rate (40% vs 0%) in the mannose group (Fig. 1c). In a type I interferon-receptor-deficient A129 mouse model for ZIKV infection, all PBS-treated mice died on or before 9dpi, whereas 40% mice in the mannose group survived (40% vs 0%, Fig. 1d). These results suggest a beneficial role of mannose as a cross-family therapy of viral diseases. Intriguingly, we found that mannose administration did not reduce SARS-CoV-2 or H1N1 viral load in mouse lungs, neither did mannose suppressed ZIKV replication in A129 mouse brain, which are the major anatomical sites of infection (Supplementary Fig. 2b–d). In cell cultures, mannose did not affect SARS-CoV-2 or H1N1 antigen expression, either (Supplementary Fig. 2e). The result prompted us to investigate the tolerance of tissue damage induced by virus infections, a critical determinant on host fitness in addition to virus burden. To this end, we performed vital sign monitoring (pulse distention, blood oxygen saturation, respiratory rate, and heart rate) in H1N1-infected mice treated with PBS or mannose (Fig. 1e). We found that mannose-treated mice maintained their pulse distention and blood oxygen saturation significantly better than that of the PBS-treated mice (Fig. 1f). To assess other evidence of end-organ damage, detailed histopathologic analysis of hematoxylin & eosin (H&E) stained sections of lung tissues was performed. Consistently, large area of consolidation with massive alveolar space infiltration and exudation were identified in animal lungs at 7dpi of H1N1 infection, whereas lung tissues in mannose treatment group exhibited improved morphology and significantly milder degree of bronchiolar infiltration (Fig. 1g). Collectively, we demonstrated the effectiveness of mannose to protect mice against infections separately caused by three species of virus from different families without suppressing the tissue viral load.

### Mannose confers protection despite delayed antiviral therapy

Clinically, patients are usually admitted to hospitals a few days after symptom onset of the viral disease. Thus, the unsatisfactory outcome of conventional antiviral therapy can be the result of delayed initiation of drug treatment, high initial viral load, and the emergence of resistance. To determine the potential of mannose as an 'universal adjuvant' to enhance the efficacy of delayed antiviral treatment, a combinatorial regimen containing a virus-targeting inhibitor and

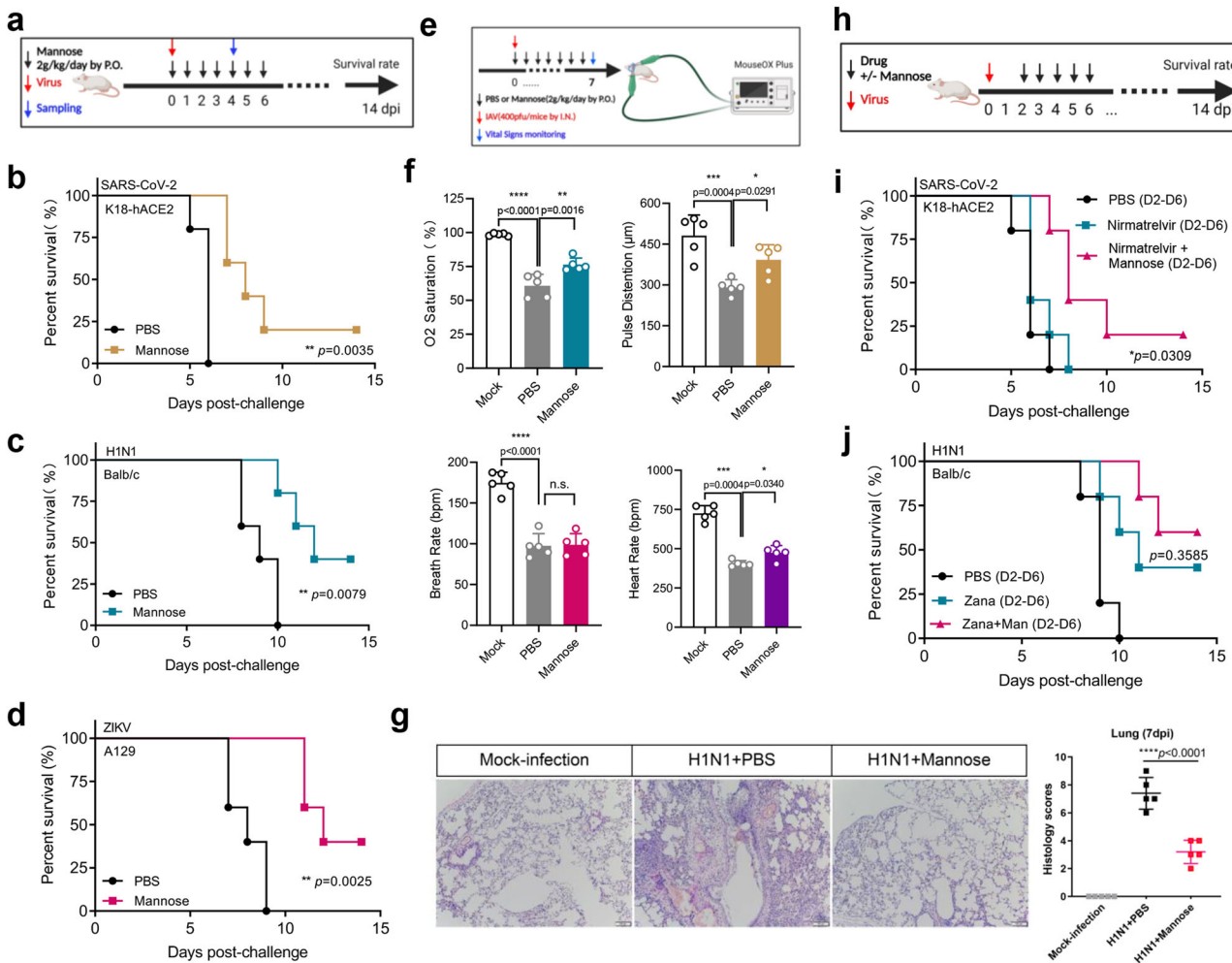

**Fig. 1 | Mannose renders protection against different viral diseases by enhancing tissue tolerance. a** Experimental design of using mannose monotherapy in three virus disease models. Mice were treated with 2 g/kg/day of mannose or PBS by oral administration (P.O.) from day0 until 6dpi. The survival rates were monitored daily until 14dpi. **b** Survival rate of SARS-CoV-2-infected K18-hACE2 transgenic mice ($2 \times 10^2$ PFU/mouse, n = 5). **c** Survival rate of mouse-adapted influenza A H1N1 virus-infected BALB/c mice ($4 \times 10^2$ PFU/mouse, n = 5). **d** Survival rate of ZIKV-infected A129 mice ($1 \times 10^6$ PFU/mouse, n = 5). **e** Schematic design showing the measurement of vital signs in H1N1-infected mice. **f** Comparison of vital signs among the indicated groups (n = 5 mice per group). **g** Representative mouse lung histopathological changes on 7dpi. Semiquantitative histology scores were given to each lung tissue (n = 5 mice per group). **h** Experimental design of using mannose combinatorial therapy in delayed treatment setting. K18-hACE2 transgenic mice were challenged with $2 \times 10^2$ PFU SARS-CoV-2 per mouse, while Balb/c mice were

challenged with $4 \times 10^2$ PFU H1N1 per mouse, at 0dpi. Virus-infected mice were treated with 2 g/kg/day of mannose from 2dpi to 6dpi, with or without antiviral-treatment (i.e., zanamivir or nirmatrelvir). The survival rates were monitored daily until 14dpi. **i** Survival rate of SARS-CoV-2-infected mice with combinational treatment of nirmatrelvir (20 mg/kg/day) and mannose (2 g/kg/day) from 2dpi to 6dpi. The mice receiving PBS treatment or nirmatrelvir alone were taken as controls (n = 5). **j** Survival rate of H1N1-infected mice with combinational treatment of zanamivir (50 mg/kg/day, I.P.) and mannose (2 g/kg/day, P.O.) from 2dpi to 6dpi. The mice receiving PBS treatment or zanamivir alone were taken as controls (n = 5). All the results are shown as mean ± SD. Comparison of survival rates between groups were analyzed using Log-rank (Mantel–Cox) tests for (**b, c, d, i, j**). One-way ANOVA with Dunnett's post hoc test was used for (**f, g**). ****$P < 0.0001$, ***$P < 0.001$, **$P < 0.01$, *$P < 0.05$, and n.s. indicates non-significant. Panels (**a, e, h**) were created with BioRender.com.

mannose were designed (Fig. 1h). In a SARS-CoV-2-infected K18-hACE2 mouse model, oral administration of mannose upon delayed treatment of Nirmatrelvir (the effective component of Paxlovid) improved the survival rate from 0% to 20%, and 2 more days of median survival (Fig. 1i). Since lethality of SARS-CoV-2 infection in K18-hACE2 mouse is caused by pneumonia and fatal encephalitis, the protective rate by mannose supplementation is expected to be higher using a more physiologically relevant mouse model. In the case of influenza A virus infection, delayed zanamivir treatment starting from 2dpi to 6dpi protected 40% survival, whereas supplementation with mannose generally improved the mouse survival to 60%, though not statistically significant (Fig. 1j). Taken together, mannose extended the therapeutic windows of effectiveness by current antiviral therapies and thus mannose may be considered as a supplementary medication for clinical management.

## Mannose rewires glycolysis and mitochondrial dysregulation

Poorly controlled high blood glucose level observed in diabetic patients is a major risk factor for severe COVID-19 or influenza which is most likely due to dysregulated cytokine production[18]. Based on the beneficial role of mannose in immunopathology[19], we hypothesized that mannose affect glucose catabolism by rewiring the imbalance in immunometabolism of the infected host (Fig. 2a). To examine, insulin was utilized during the infection course (Fig. 2b). At 1dpi and 3dpi, we found that H1N1 infection consistently reduced blood glucose level, indicating a heightened glycolysis demand; whereas mannose administration normalized the blood glucose level to the similar level of non-infected mice (Fig. 2c). Concordantly, co-application of insulin and mannose abolished the protective role of mannose (0% vs 40%) with a median survival duration even less than the PBS control (7days vs 9days) (Fig. 2c). These results suggest that mannose reduces cellular

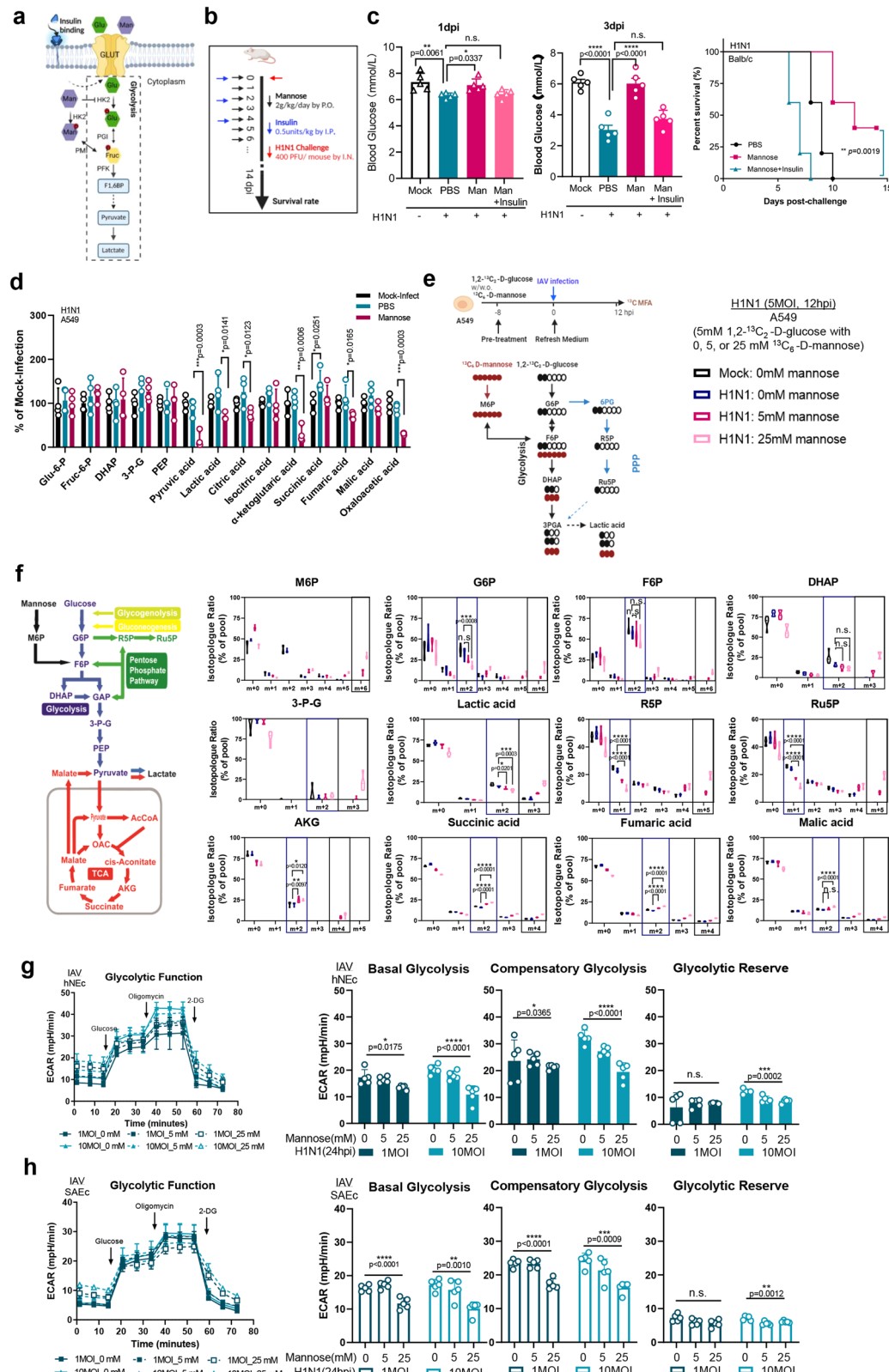

utilization of glucose in order to exert host protection. To view the landscape of metabolic change after mannose treatment, we performed a targeted metabolomics analysis with a focus on glucose homeostasis. Notably, we found marked increase of 3-phosphoglycerate and lactic acid after H1N1 infection in human lung epithelial A549 cells, which were reversed after mannose treatment. In the TCA cycle which is downstream of glycolysis, a similar trend of

metabolic changes was documented (Fig. 2d). To further confirm if mannose inhibits glucose- derived glycolytic intermediates, we conducted a $^{13}$C-labeled metabolic flux analysis (MFA) using 1,2-$^{13}C_2$-glucose and $^{13}C_6$-mannose labeled cells, which differentiate glucose from mannose (Fig. 2e). Indeed, the mannose-treated and H1N1-infected A549 cells exhibited generally lower levels of $^{13}C_2$-glucose-6-phosphate (G6P), $^{13}C_2$-fructose-6-phosphate (F6P), and $^{13}C_2$-lactic acid (Fig. 2f).

**Fig. 2 | Mannose affects host glucose metabolism and mitochondrial respiration. a** Schematic illustration of the roles of mannose (Man) and glucose (Glu) in the glycolysis pathway. Mannose enters glycolysis with the common GLUT family and acts as competitor of glucose for hexokinase (HK) enzyme HK2, first being converted to mannose-6-phosphate that is then isomerized to produce fructose-6-phosphate by Mannose-6-phosphate isomerase (PMI) enzyme. **b** Experimental design showing the co-administration of insulin and mannose after H1N1 infection. **c** Left panel: measurement of blood glucose of H1N1-infected BALB/c mice ($n = 5$ mice per group) at 1dpi and 3dpi, respectively. Right panel: survival rate of H1N1-infected and mannose-treated BALB/c mice with or without insulin administration (0.5 units/kg, I.P.). **d** Metabolomic profile of H1N1-infected human lung epithelial A549 cell after mannose treatment ($n = 4$ biological repeats). A549 cells were pre-treated with mannose (25 mM) or PBS overnight, followed by H1N1 infection (2MOI) for another 12 h before targeted metabolomics analysis of glycolysis and tricarboxylic acid (TCA) cycle. The results were normalized by cell number. **e** Schematic of 1,2-$^{13}C_2$-D-glucose and $^{13}C_6$-D-mannose labeling of carbon atoms in glycolysis intermediates arising from glucose metabolism, pentose phosphate pathway (PPP), and TCA cycle. **f** Extraction of intracellular metabolites and measurement of their mass isotopologue distribution (MID). Isotopologue ratio of the indicated metabolite was measured and normalized to the pool of relevant metabolite, respectively ($n = 5$ biological repeats). **g** Human primary nasal epithelial cells (hNEc) and (**h**) human primary Small Airway Epithelial cells (SAEc) were pre-treated with 0 mM, 5 mM, or 25 mM mannose overnight before virus infection for another 12 h. Cells were analyzed using a Seahorse XF analyzer to determine the extracellular acidification rate (ECAR). The ECAR data were normalized using Hoechst 33342 nucleic acid stain to quantify the number of live cells ($n = 5$ biological repeats). All the results are shown as mean ± SD. Comparison of survival rates between groups were analyzed using Log-rank (Mantel–Cox) tests for (**c**). One-way ANOVA with Dunnett's post hoc test was used for (**c, f, g, h**). ****$P < 0.0001$, ***$P < 0.001$, **$P < 0.01$, *$P < 0.05$, and n.s. indicates non-significant. Panels (**a, b, e, f**) were created with BioRender.com.

Moreover, we measured the extracellular acidification rate (ECAR) as a real-time analysis of glycolysis. Three H1N1-permissive cell types were examined, including human Small Airway Epithelial Cells (SAEc), human Nasal Epithelial Cells (hNEc) and A549 cells. Consistently, we found that H1N1-infection elevated glycolytic rate in all the three cell types, whereas mannose treatment significantly lowers the glycolytic function (Fig. 2g, h, Supplementary Fig. 3a, b).

Mitochondria are well appreciated as biosynthetic and bioenergetic organelles for their roles in producing metabolites and ATP, which are products of the TCA cycle and the mitochondrial membrane potential, respectively. To determine if mannose affects the TCA intermediates, both metabolic flux analysis and oxygen consumption rate measurement were conducted using the same experimental settings as that for glycolysis analysis. We found that mannose-treated groups exhibited increased ratio of glucose-derived alpha-ketoglutarate (AKG), succinic acid, fumaric acid, and malic acid, indicating a reprograming of TCA cycle and pentose phosphate pathway (Fig. 2f). In line with this observation, Oxygen Consumption Rate (OCR) analysis reveals that mannose treatment increases respiration rate in the mitochondria of both SAEc and hNEc (Figs. 3a, b). Surprisingly, a reverse OCR pattern was detected in A549 cells (Supplementary Fig. 3c, d), indicating that mannose may not only affect glucose metabolism, but also utilization of other carbon sources (e.g. glutamine, amino acids, urine etc.). Nevertheless, we confirmed that mannose rewires virus-induced glycolysis and TCA dysregulation as a mode of host protection.

## Mannose antagonizes the overwhelming host inflammatory response

The roles of mannose in inducing regulatory T cells and suppressing immunopathology have been reported[20]. Torretta and colleagues have also documented that mannose can suppress inflammatory macrophage activation by impairing IL-1β production[10]. Utilizing human peripheral blood mononuclear cells, we found that mannose treatment increased B cell population, which may partially explain the beneficial effects of mannose (Fig. 3c). In addition, we performed transcriptomic analysis in A549 cells and demonstrated a reduced proinflammatory response but increased host tolerance upon H1N1 virus infection and mannose therapy (Fig. 3d and Supplementary Fig. 4), providing further evidence for the beneficial role of mannose in immunometabolism. Because the capacity of glycolysis to induce proinflammatory cytokines production has been recognized[10], together with the phenotypic changes of metabolites observed in this study, we hypothesized that succinate might be one of the critical metabolites reversing the proinflammatory cytokine production by mannose (Fig. 3e). Indeed, we found that succinic acid (succinate) was upregulated upon H1N1 infection and suppressed after mannose treatment (Fig. 3f). Since the levels of succinate affect HIF-1α activity, a key transcription factor in the expression of pro-inflammatory genes,

we monitored the protein expression level of HIF-1α in the context of virus infection and mannose treatment. As shown in Fig. 3g, increased HIF-1α expression was detected in both SARS-CoV-2 infected Calu3 cells and H1N1-infected A549 cells, but the amounts were consistently suppressed after treatment with mannose. Because succinate supports the tricarboxylic acid cycle and ATP generation in mitochondria, we measured the mitochondrial membrane potential upon virus infection with or without mannose treatment. JC-1, a cationic carbocyanine dye, that accumulates in mitochondria was used. In H1N1 infected cells, we found a substantial depolarization of the inner mitochondrial membrane, which is reflected by the averaged monomer (Green) signal of JC-1 from 9.87% (mock group) to 68.25% (H1N1 group) in A549 cells (Fig. 3h). As expected, administration of mannose reduced the inner mitochondrial membrane depolarization induced by H1N1 infection. Exploring along this axis, we next determined the in vivo effect of inflammatory control after reprogramming glycolysis by mannose (Fig. 3i). In BALB/c mice intranasally challenged with H1N1, mannose halved the amount of IL-1β in bronchoalveolar lavage (BAL) fluid (Fig. 3j). Systematic administration of mannose also reduced the succinate production in mouse lung (Fig. 3k). Therefore, mannose reverses the virus-induced inflammatory dysregulation via the succinate-HIF-1α axis. Moreover, decreased ROS were detectable in the lung tissue of H1N1-infected BALB/c mice after mannose administration (Fig. 3l). Taken together, we concluded that mannose elevates host tissue tolerance to damage by virus infection via antagonizing the HIF-1α-mediated proinflammatory response.

## The beneficial role of mannose is PMI-dependent

Within the glycolysis pathway, Fructose-6-phosphate (F6P) is isomerized either from the M6P via PMI or G6P via Glucose-6-phosphate isomerase (PGI), followed by the rate-limiting control of phosphofructokinase (PFK) (Fig. 2a). Since intracellular M6P exists predominantly after outcompeting G6P, we reasoned that the beneficial role of mannose may be connected to the abundance of the M6P catalyzing enzyme PMI. To this end, we utilized a *PMI* knockout human cell line A549-*PMI*$^{-/-}$ for the evaluation of the outcome after virus infection with or without mannose treatment. First, we examined the viability of *PMI*$^{-/-}$ cells by the cell proliferation assay. We found that *PMI*$^{-/-}$ cells exhibited comparable growth rates when compared to the wild-type A549 cells (Fig. 4a). Mannose treatment generally does not suppress cell proliferation nor impair cell viability as examined in hNEc, SAEc and A549 cells (Supplementary Fig. 5). With mannose addition, unexpectedly, retardation of cell growth was observed in *PMI*$^{-/-}$ cells but not in WT cells. This result indicates an unfavorable effect upon the combination of mannose treatment and PMI depletion. Apparently, metabolic profiling revealed that *PMI* knockout decreased both glycolytic rate and mitochondrial respiration after H1N1 infection, which explains the observation that simultaneous suppression of

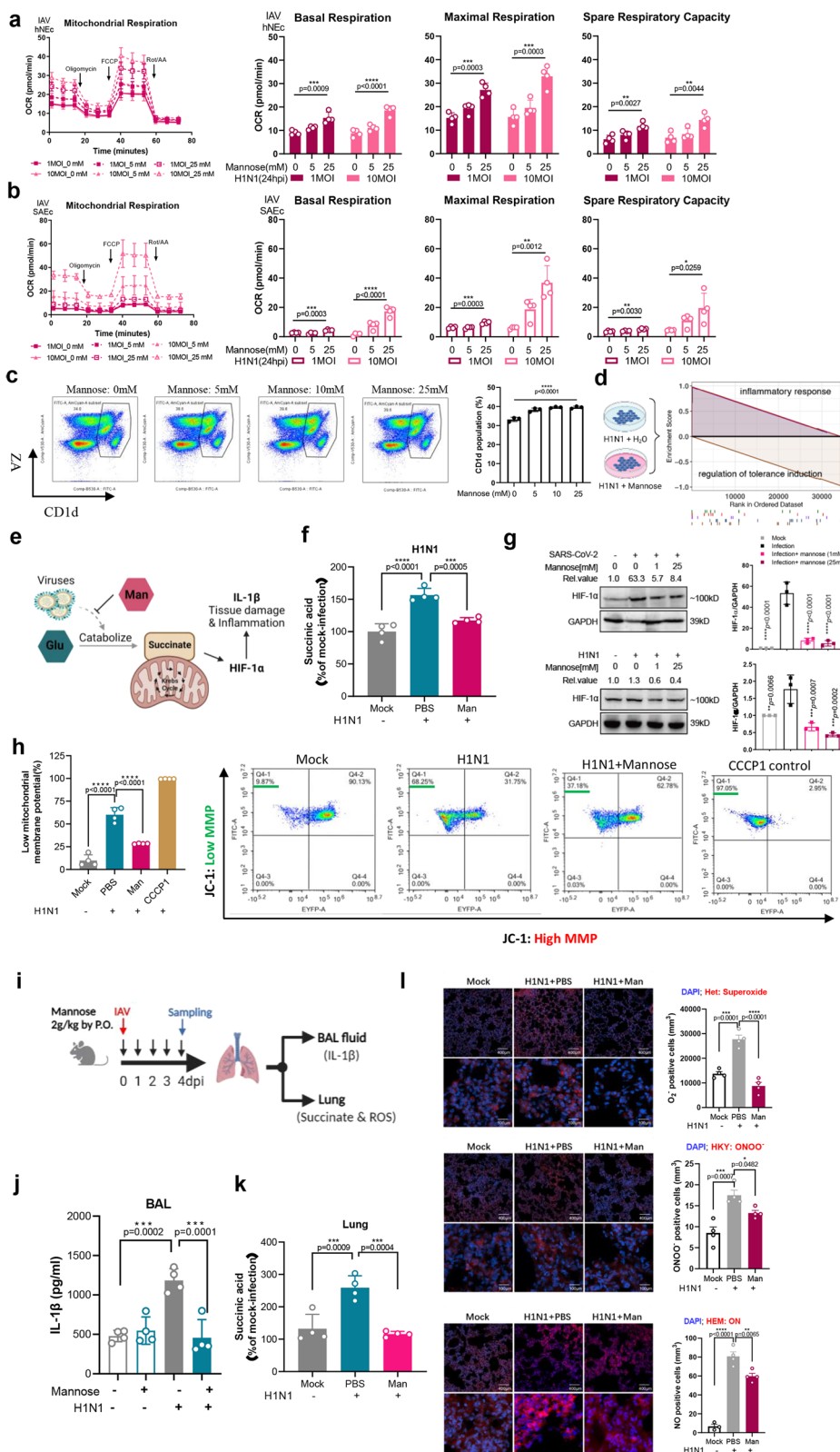

glucose influx and PMI depletion may completely shut down host glycolysis and energy supply, thus impairing cell proliferation (Figs. 4b, c and Supplementary Fig. 6).

To confirm the critical role of PMI on mannose metabolism, we employed ectopic overexpression of PMI and/or supplementation with the PMI metabolite F6P for a potential change of phenotype. We found that PMI overexpression(O/E) or F6P supplementation or both partially

abolished the ameliorative function of mannose on mitochondrial damage upon virus infection (Fig. 4d). In terms of PMI overexpression, either it fuels more virus production via glycosylation, or it facilitates the transfer of D-mannose to D-fructose which rescued the glycolysis suppressed by mannose. However, PMI depletion abrogated the benefits of mannose to rewire mitochondrial dysfunction and induced even higher mitochondrial membrane potential than that of the virus control group

**Fig. 3 | Mannose rewires virus-induced mitochondrial dysfunction and inflammatory dysregulation. a** Human primary hNEc and (**b**) SAEc cells were analyzed the oxygen consumption rates of mitochondria after H1N1 infection and mannose treatment ($n$ = 4 biological repeats). **c** Human PBMCs were treated with mannose in the indicated concentration for overnight. After treatment, cells were collected and stained with fluorescently labeled antibodies specific for B cell surface markers CD1d before flow cytometry analysis ($n$ = 3 independent donors). **d** Transcriptomic analysis of H1N1-infected A549 cells suggests decreased inflammatory response whereas enhanced tolerance induction after mannose treatment. Gene set enrichment analysis enrichment plots of the most significant target classes are shown. **e** Schematic illustration showing the glycolysis/succinate/HIF-1α/ IL-1β axis that mannose interferes with. **f** The succinic acid level in the supernatant of H1N1-infected cells were determined ($n$ = 4 biological repeats). **g** The HIF-1α protein expression after mannose treatment was assessed by western blotting in SARS-CoV-2-infected Calu3 cells (0.1 MOI, 24hpi) and H1N1-infected A549 cells (0.1 MOI, 24hpi), respectively. **h** The mitochondrial membrane potential of H1N1-infected cells were analyzed by flow cytometry after JC-1 dye staining ($n$ = 4 biological repeats). **i** Experimental design showing the investigation of in vivo inflammation response after oral mannose treatment. H1N1 (400PFU) infected and mannose-treated BALB/c mice were sacrificed on 4dpi for downstream analysis. **j** The IL-1β amount in the bronchoalveolar lavage (BAL) fluid of mice ($n$ = 4 mice per group) were determined by ELISA. **k** The amount of succinic acid in mouse lung homogenates ($n$ = 4 mice per group) were detected by ELISA. **l** The reactive oxygen species (ROS) intensity in the mouse lung were stained with three specific dyes for superoxide. Shown are representative images randomly selected from a pool of images for each group ($n$ = 4 mice/group). The results are qualified as $O_2^-$, $ONOO^-$, and ON positive cells per $mm^3$ in four randomly selected areas, respectively. All the results are shown as mean ± SD. One-way ANOVA with Dunnett's post hoc test was used for (**a, b, c, f, g, h, j, k, l**). ****$P < 0.0001$, ***$P < 0.001$, **$P < 0.01$, *$P < 0.05$, and n.s. indicates non-significant. Left part of (**d**), and (**e**), and (**i**) were created with BioRender.com.

(Fig. 4e). Of note, mannose increased the damage to mitochondrial membrane potential in $PMI^{-/-}$ cells. However, addition of F6P upon $PMI^{-/-}$ and mannose treatment restored the protective function of mannose (Fig. 4e). Utilizing TMRE staining on both hBTEC and A549 cell models, we further confirmed the reduced mitochondrial membrane potential caused by virus infection, which was protected by mannose whereas antagonized by F6P (Fig. 4f). The result not only ascertained the importance of PMI in enabling mannose to antagonize against virus-induced mitochondrial abnormity, but also affirmed that heightened glycolysis dysregulates mitochondrial membrane potential. Along this direction, we further confirmed that mannose treatment in A549-WT cells suppressed IL-1β secretion upon poly(I:C) transfection, whereas PMI overexpression rescued IL-1β production in the presence of mannose (Fig. 4g). In A549-$PMI^{-/-}$ cells, however, mannose treatment and PMI overexpression exhibited an adverse phenotype when compared with that of the WT group (Fig. 4g). Taken together, PMI is critical in determining the detrimental or beneficial role of mannose during glycolysis reprogramming.

## PMI is an antiviral target that affect protein glycosylation

To understand the host PMI response to virus infection, we determined its mRNA level in cells infected by SARS-CoV-2, H1N1, ZIKV, and EV-A71, respectively. Since $PMI$ mRNA expression increased steadily post-infection, PMI inhibition may serve as an antiviral strategy (Fig. 5a). Indeed, siRNA-mediated $PMI$ gene silencing suppressed virus replication in H1N1 (-0.5 log), SARS-CoV-2 (-1 log) and ZIKV (-0.8 log), individually (Fig. 5b). Moreover, pharmacological inhibition using a reported PMI inhibitor, MLS0315771, exhibited antiviral effect against the same panel of viruses at non-toxic concentrations, with an $EC_{50}$ of 3.08 μM against SARS-CoV-2, 3.17 μM against H1N1, and 4.37 μM against ZIKV (Fig. 5c). To explore which step of virus life cycle that PMI is responsible for, we generated vesicular stomatitis virus (VSV)-based spike pseudotyped viruses for determination of virus entry efficiency (Strategy 1, Fig. 5d). As shown in Fig. 5e, both $PMI$ knockout and MLS0315771 treatment reduced chemiluminescence signal, indicating that PMI contributes to SARS-CoV-2 entry. However, mannose did not affect virus entry or subsequent virus replication cycle, nor did mannose treatment reduce the PMI expression (Supplementary Fig. 2e). These results suggest that PMI is a versatile molecule that functions not only as a gatekeeper of mannose-mediated glycolysis reprogramming, but also other parts of glucose metabolism which affect virus replication fitness.

Since PMI is an essential enzyme in the early steps of the protein glycosylation pathway, and inhibition of glycosylation with tunicamycin was reported to block SARS-CoV-2 infection[21]. We hypothesized that PMI may affect N-glycosylation of either host or viral proteins or both to interfere with virus entry. To validate if PMI affect Spike glycosylation, we packaged the VSV-based SARS-CoV-2-spike

pseudotyped viruses in WT or $PMI^{-/-}$ or $PMI^{-/-}$ cells rescued with exogeneous PMI (Strategy 2, Fig. 5d). We demonstrated that $PMI^{-/-}$ significantly impaired the pseudovirus entry into cells by both ancestral and Omicron BA.5 spike-pseudotyped virus, whereas exogenous supply of PMI rescued the virus entry efficiency (Fig. 5f). Moreover, the PMI inhibitor MLS0315771 added during pseudovirus packaging compromised the resultant entry efficiency, too (Fig. 5g). These results indicate that PMI may play a role in spike glycosylation to facilitate SARS-CoV-2 infection. Utilizing an established lectin blot analysis in tandem with Western blot analysis[22], indeed, addition of MLS0315771 on spike protein overexpression resulted in dose-dependent reduction of spike glycosylation (Supplementary Fig. 7a), which partially explains the antiviral mode of MLS0315771. To validate if PMI influences host Angiotensin-converting enzyme 2 (hACE2) glycosylation, we performed the similar analysis of this SARS-CoV-2 receptor, which harbors six sites of N-glycan microheterogeneity with an important role in molecular dynamic interaction between Spike and ACE2[23,24]. We found impaired hACE2 glycosylation in $PMI^{-/-}$ cells, showing smear bands around 125 kDa in both input and pull-down groups (Fig. 5h). When compared with that of the WT, the relative high mannose-type N-glycans (HHL) level per amount of hACE2 in $PMI^{-/-}$ group dropped by 50%. As expected, treatment using 100 ng/ml Tunicamycin caused 70% reduction of hACE2 glycosylation, whereas overexpression of PMI in $PMI^{-/-}$ cells partially restored the HHL level of hACE2 by 20% (Fig. 5h). In a HEK293T cell model, we found global upregulation of host glycosylation process after PMI overexpression, where 750 ng and 1500 ng PMI overexpression increased the relative HHL level of hACE2 by 120% and 80%, respectively (Fig. 5i). Consistently, PMI inhibition by the compound MLS0315771 suppressed hACE2 glycosylation (Fig. 5j). These results indicate that PMI is critical for both SARS-CoV-2 spike and hACE2 glycosylation, which explains the SARS-CoV-2 entry deficiency after PMI inhibition/depletion (Fig. 5e). Importantly, we found this mode of action has broad relevance to other respiratory viruses. Utilizing H1N1 as an example, we found diminished viral hemagglutinin (HA) glycosylation in $PMI^{-/-}$ cells, whereas PMI overexpression reversed the glycosylation deficiency on HA protein (Supplementary Fig. 7b). The percentage of H1N1-NP positive cells reduced from 3.8% to 0.2% after PMI depletion (an 18-fold reduction, Supplementary Fig. 7c). Therefore, PMI may broadly mediate different virus entry machinery via glycosylation of host and/or viral surface proteins. Taken all together, we demonstrate the versatility of PMI to dictate mannose-reprogrammed glycolysis for enhanced tissue tolerance, as well as to modulate virus entry for potential antiviral drug development.

## Discussion
In this study, we outlined a previously unrecognized ability of mannose in maintaining tissue tolerance against virus infection, which is PMI dependent. Inflammatory responses must be coupled to specific

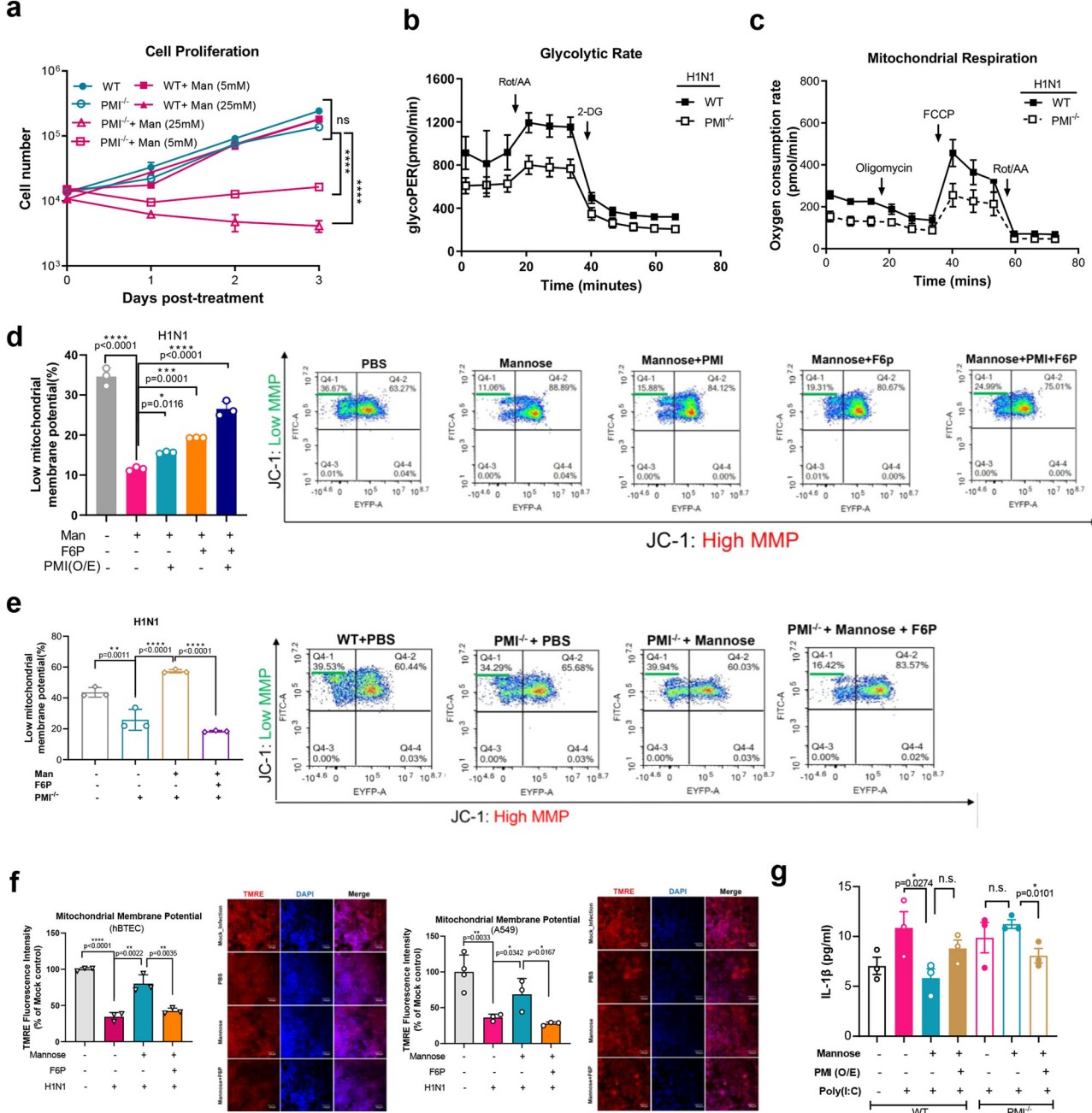

**Fig. 4 | The beneficial effect of mannose is PMI dependent. a** Growth curves of A549-wildtype (WT) and knockout (*PMI^-/-*) cells supplemented with 0, 5, or 25 mM of mannose (*n* = 4 biological repeats). **b** Bioenergetic profiles of H1N1-infected WT and *PMI^-/-* cells supplemented with or without mannose (25 mM) treatment. Cells were pre-treated with mannose overnight before virus infection for another 12 h. Glycolytic rate of the cells were analyzed using a Seahorse XF analyzer by measuring the glycoPER kit (*n* = 5 biological repeats). **c** Mitochondrial respiration of the cells was analyzed using a Seahorse XF analyzer by measuring the oxygen consumption rates (OCR, *n* = 5 biological repeats). **d** Either PMI overexpression (O/E) or supplement of PMI metabolite Fructose-6-phosphate (F6P) antagonize the suppression of low mitochondrial membrane potential (MMP) by mannose. A549 cells were pre-treated with the indicated treatment for overnight before H1N1 infection (MOI = 2). After 12 h, cells were subject to MMP analysis after JC-1 staining (*n* = 3 biological repeats). **e** Dual PMI depletion and mannose treatment is unfavorable whereas

supplement of F6P reverses the mitochondrial damage. A549-WT and *PMI^-/-* cells were pre-treated with the indicated treatment, followed by H1N1 infection and MMP measurement after JC-1 staining (*n* = 3 biological repeats). **f** MMP assay by TMRE Staining. hBTEC (left panel) and A549 cells (right panel) were pre-incubated with the indicated treatment for 12 h before H1N1 virus infection (MOI = 0.2). Cells were subject to TMRE (200 nM) staining for 20 min, and DAPI staining for normalization. The fluorescence intensities were detected by a plate reader, whereas the images were captured by GE IN Cell Analyzer 6500HS (*n* = 3 biological repeats). **g** Knockout of *PMI* impair the modulatory activity of mannose against IL-1β production. The experiments were performed in WT and *PMI^-/-* cells transfected with PMI plasmid and/or Poly(I:C) as indicated (*n* = 3 biological repeats). All the results are shown as mean ± SD. One-way ANOVA with Dunnett's post hoc test was used for (**a, d, e**). Unpaired and two-sided student's *T* test was used for (**f, g**). ****$P < 0.0001$, ***$P < 0.001$, **$P < 0.01$, *$P < 0.05$, and n.s. indicates non-significant.

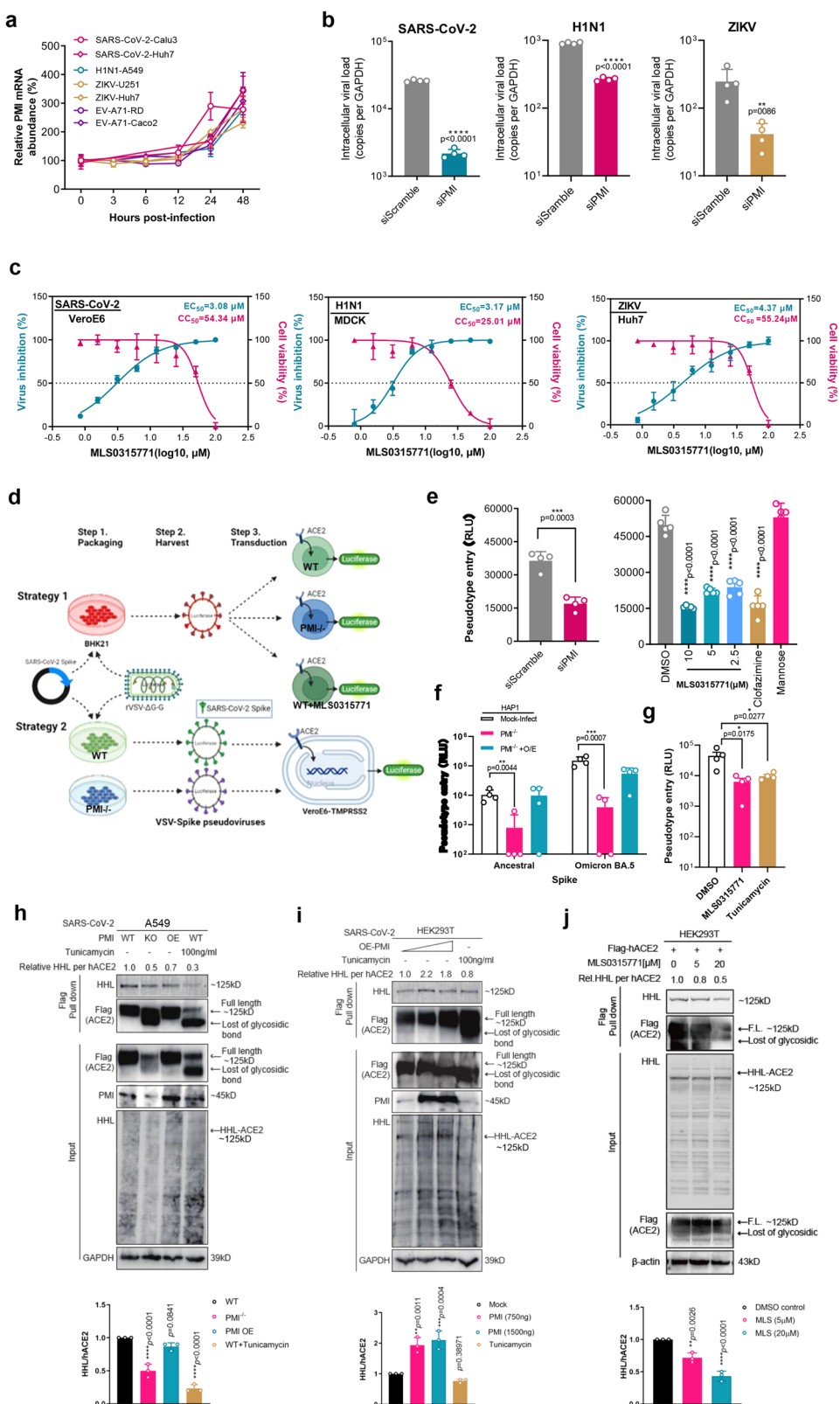

metabolic programs to support their energy demands[25], whereas immunometabolic reprogramming by mannose leads to the reduction of collateral damage. Indeed, compelling evidences have shown that controlled caloric restriction (CR), instead of leading to malnutrition, reduces inflammatory responses, autoimmunity and cancer[26,27]. We found that PMI activity dictates the beneficial role of mannose after virus infection which sustains the physiological energy supply despite

the blocking of glycolysis by mannose for the control of inflammatory damage (Fig. 6). As a result of competition from mannose, the accumulation of M6P by HK2 enzyme inhibits hexokinase (70%), phosphoglucose isomerase (65%), and glucose-6-phosphate dehydrogenase (85%) metabolism[28]. This inhibition limits the flux through glycolysis and the expression of the genes involved in glycolysis. Therefore, pharmacological inhibition or genetic knockout of *PMI*, together with mannose

**Fig. 5 | PMI affects virus entry via modulation of protein glycosylation. a** Time-dependent monitoring of *PMI* gene expression after virus infection. PMI mRNA was determined by RT-qPCR in indicated virus-infected cell models (all with 1MOI) ($n = 4$ biological repeats). **b** siRNA knockdown of *PMI* decreased viral replication in H1N1-infected A549 cells (0.1MOI, 24hpi), SARS-CoV-2-infected Caco2 cells (0.1MOI, 24hpi), and ZIKV-infected Huh7 cells (0.1MOI, 48hpi) ($n = 4$ biological repeats). **c** The PMI inhibitor MLS0315771 inhibits the replication of a panel of viruses including H1N1-infected MDCK cells (0.01 MOI, 24hpi), SARS-CoV-2-infected VeroE6 cells (0.01 MOI, 24hpi), and ZIKV-infected Huh7 cells (0.1 MOI, 48hpi) ($n = 3$ biological repeats). **d** Schematic illustration showing two strategies of SARS-CoV-2 spike-mediated entry measurement. **e** PMI inhibition either by siRNA knockdown or by inhibitor MLS0315771 reduced SARS-CoV-2 pseudovirus entry in HEK293T-ACE2 cells (strategy 1) ($n = 4$ biological repeats). **f** Overexpression (O/E) of PMI rescued SARS-CoV-2 entry (strategy 2). Both WT and Omicron BA.5 pseudovirus were examined ($n = 4$ biological repeats). **g** Addition of Tunicamycin (2 μg/ml) or MLS0315771 (5 μM) to HEK293T cells during packaging of pesudotyped SARS-

CoV-2 decreased its entry efficacy VeroE6-TMPRSS2 cells (strategy 2) ($n = 4$ biological repeats). **h** PMI mediates high mannose-type N-glycosidic (HHL) glycosylation of host ACE2. After SARS-CoV-2 infection (0.1MOI, 24hpi) and pull-down experiment, full spectrum of host HHL glycosylation process particularly hACE2 were analyzed by Western blot and lectin blot analyses ($n = 3$ biological repeats). **i** Overexpression of PMI increased the hACE2 glycosylation. HEK293T cells were transfected with increasing concentrations of exogenous PMI plasmids followed by SARS-CoV-2 infection and analysis of hACE2 N- glycosylation as described in (**h**). **j** The PMI inhibitor MLS0315771 decreased the hACE2 glycosylation. HEK293T cells were transfected with hACE2 constructs and treated with MLS0315771 for 24 h before analysis of hACE2 N- glycosylation as described in (**h**) ($n = 3$ biological repeats). All the results are shown as mean ± SD. Unpaired two-sided student's *T* test was used for figure (**b**, **e**) (left panel). One-way ANOVA with Dunnett's post hoc test was used for (**e**) (right part), (**f, g, h, i, j**). ****$P < 0.0001$, ***$P < 0.001$, **$P < 0.01$, *$P < 0.05$, and n.s. indicates non-significant. Panel (**d**) was created with BioRender.com.

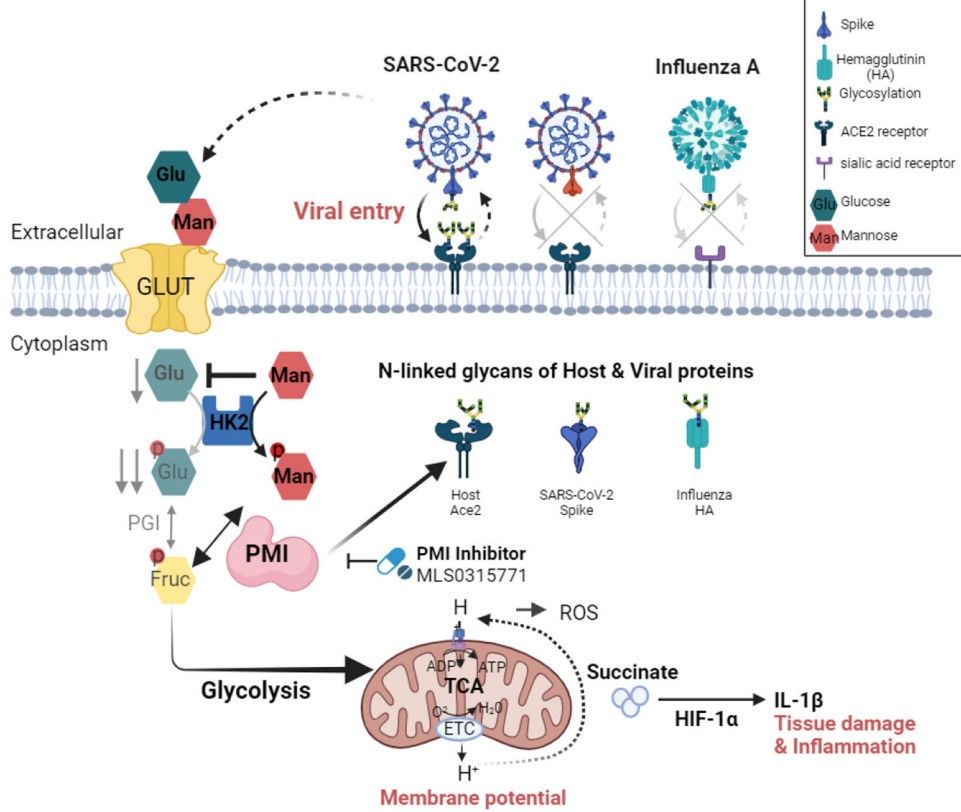

**Fig. 6 | Proposed model of PMI-mediated glycolysis reprogramming and glycosylation to affect tissue tolerance and virus fitness.** Virus infection triggers heightened energetic demand including increased glucose influx and upregulated glycolysis. Mannose antagonizes such process by competing with glucose for glucose translocation across the cell membrane (GLUT), as well as for hexokinase enzyme (HK2) that are critically required during glycolysis. Consequently, this metabolic reprogramming by mannose normalizes the mitochondrial dysfunction as evidenced

by heightened membrane potential and reactive oxygen species. It also reverses the abnormally high production of succinate from TCA cycle, HIF-1α activation and induction of overwhelming IL-1β that causing tissue damage. PMI is a gatekeeper that determines it is beneficial or unfavorable during glycolysis remodeling after mannose supplement. PMI can be also taken as a druggable target that affects a panel of virus entry. Mechanistically, PMI contributes to both host ACE2 and virus spike glycosylation thus controls SARS-CoV-2 entry. Created with BioRender.com.

supplement, resulted in impaired cell proliferation (Fig. 4a) and mitochondrial health (Fig. 4e). With PMI activity maintained, however, mannose therapy can be taken as a safe and therapeutic option without the complete shutdown of host glyco-energetic supply. Instead, mannose rewires the virus-induced hyper-glycolysis, and normalized the downstream mitochondrial respiration and inflammatory response (Fig. 6). This mode of action is clinically significant, as the arrest of inflammatory tissue damage is particularly important in outcome of viral diseases, if not more essential than the regular antiviral monotherapy and particularly in a delayed treatment scenario.

PMI is a housekeeping enzyme found in organisms ranging from bacteria to fungi to mammals which is important for cell-wall synthesis, viability and signaling. Based on their amino acid sequence identity, PMIs have been classified into three unrelated families[29]. Type I PMIs are zinc-dependent monofunctional aldose-ketose isomerases and include all eukaryotic and some bacterial enzymes. As Type I PMIs have a high level of sequence identity, particularly in the region of the active site[29], promising clinical implications are achievable leveraged on the encouraging animal data in the present study. PMI interconverts F6P and M6P, therefore linking glycolysis to protein glycosylation. M6P is

further converted to M1P by phosphomannose mutase (PMM) followed by production of GDP-Man, which is the central activated mannose donor in glycosylation reactions. Normally, PMI provides the majority of mannose for glycan synthesis. Loss of PMI decreases the GDP-mannose pools and limits the amount of available lipid-linked oligosaccharide that can bind to proteins. Indeed, we found decreased glycosylation of SARS-CoV-2 spike in $PMI^{-/-}$ cells, which are critical to mediate virus entry (Fig. 5f). Moreover, PMI contributes to H1N1 entry, partially via glycosylation of the viral hemagglutinin (HA) protein (Supplementary Fig. 7). Because PMI ablation is embryonic lethal[11], we were unable to examine its influence on multiple viruses' replication in vivo. Nevertheless, our findings uncover a previously unreported role of PMI to promote post-translational modification of a panel of medically important viruses.

PMI can be a druggable target in the absence of mannose supplement. Changes in viral protein glycosylation can affect interaction with receptors and may render a virus to be more recognizable by the host innate immune effectors and less recognizable by antibody, thus impacting viral entry, replication, and infectivity[30]. Many pathogenic viruses including SARS-CoV-2, H1N1, HIV, and West Nile virus require glycosylation of their proteins to achieve their pathogenesis and immune evasion. For example, SARS-CoV-2 genome encodes lots of highly glycosylated proteins, such as spike, envelope, membrane, and ORF3a proteins, which are responsible for host recognition, penetration, binding, recycling and pathogenesis. Influenza HA undergoes post-translational host-cell dependent glycosylation that are crucial to its proper folding and trafficking during infection. Taken together, PMI may serve as a host based broad-spectrum antiviral target.

Our findings may have important clinical implications. Though medium dose dexamethasone, IL6 antagonist such as tocilizumab, JAK kinase inhibitor such as baricitinib are recommended to reduce the inflammatory damage for severe COVID-19, their broad spectrum of immunosuppression also predispose patients to other infections[31]. The use of oral mannose to improve tissue tolerance and reduce inflammation may be an alternative strategy to improve the outcome of severe or lately presented viral infections, in addition to the use of an effective antiviral agent. The physiological level of D-mannose in the blood of humans is approximately 100 μM. In humans, stable serum D-mannose levels of up to 2 mM can be reached and are well tolerated, without signs of liver or renal toxicity[32]. In contrast to its effects on glucose metabolism, mannose does not decrease the uptake of amino acids or fatty acids. Though mannose reduces glucose-dependent serine and glycine synthesis, this contributes only marginally to total cellular serine and glycine pools. Moreover, mannose is biologically active and widely accessible in fruits (e.g., cranberries) and plants. Nutritional supplementation of mannose thus holds the potential as a universal adjuvant during clinical management of infectious diseases. This is particularly important for critical illness when admitted patients are likely to suffer from both high virus burden and tissue inflammatory damage which considerably compromise the efficacy of existing antiviral agents, whereas mannose supplements extend the therapeutic window for better clinical management. In sum, our findings warrant further exploration of the basic immunological mechanisms of metabolic rewiring and the potential clinical applications of hexose sugars.

## Methods

### Cells and viruses

The Human Small Airway Epithelial Cells (SAEc) were obtained from the Lonza company (Catalog #: CC-2547) and were cultured using the specialized Small Airway Epithelial Cell Growth Medium (Catalog #: CC-3119). The Human Nasal Epithelial Cells (hNEc) were obtained from the promocell company (Catalog #: C-12620) and were cultured using the specialized Airway Epithelial Cell Growth Medium (Catalog #: C-21060). The Bronchial/Tracheal Epithelial Cells (hBTEc) were obtained from the ATCC company (ATCC PCS-300-010) and were cultured with ATCC complete growth media, consisting of ATCC Airway Epithelial

Cell Basal Medium (ATCC PCS-300-030) supplemented with Bronchial Epithelial Cell Growth Kit (ATCC PCS-300-040). The human embryonic kidney-derived cell line that constitutively expresses the human angiotensin I converting enzyme 2 cells (293T-ACE2), human lung carcinoma A549 and Calu-3 cells, human hepatoma Huh7 cells, human rhabdomyosarcoma RD cells, human glioblastoma U251 cells, Madin-Darby canine kidney cells (MDCK), and African green monkey kidney Vero and VeroE6 cells were maintained in DMEM medium. The $PMI$ knockout cells (A549-$PMI^{-/-}$) were constructed by CRISPR genome editing and grown in DMEM medium. All culture medium was supplemented with 10% heat-inactivated FBS, 50U/ml penicillin, and 50 μg/ml streptomycin. All cells were confirmed to be free of mycoplasma contamination by Plasmo Test (InvivoGen). The mouse-adapted influenza A virus strain A/Hong Kong/415742/2009(H1N1)pdm09 was cultured in embryonated chicken eggs. The SARS-CoV-2 strain B.1.1.7/Alpha (GenBank: OM212469) was propagated in VeroE6 cells. Enterovirus A-71 (SZ/HK08-5) was cultured in RD cells. Zika virus (Puerto Rico strain PRVABC59, a gift from Dr. Brandy Russell and Dr. Barbara Johnson, CDC, USA) was amplified in VeroE6 cells. All cultured viruses were titrated by plaque-forming unit (PFU) assays and/or 50% tissue culture infectious dose (TCID$_{50}$) assay and/or RT-qPCR assays as indicated. All virus stocks were kept at −80 °C in aliquots. All experiments with live viruses were conducted using biosafety level 2 or 3 facilities as we previously described[33,34].

### Antiviral assessment in different mouse models

Male and female mice were kept in biosafety level 2 or 3 housing and given access to standard pellet feed and water ad libitum, with individual ventilation with 65% humidity and ambient temperature ranging between 21 and 23 Degree Celsius with 12-hour-interval day/night cycle for housing and husbandry[35]. All experimental protocols were approved by the Animal Ethics Committee (CULATAR) in the University of Hong Kong and were performed according to the standard operating procedures biosafety level 2 animal facilities. Male K18-hACE2 transgenic mice[36], female BALB/c mice, and female IFNα/βR$^{-/-}$ (A129) mice were bred, raised in the HKU Center for Comparative Medicine Research and delivered upon reaching 6−8 weeks of age. After anesthesia, mice were intranasally (i.n.) inoculated with 20 μl of virus suspension containing 200 PFU of SARS-CoV-2, or 400 PFU of H1N1, or 10,000 PFU of ZIKV, respectively. Therapeutic treatments with mannose (2 g/kg/day, BID, P.O.), or zanamivir (50 mg/kg/day, QD, I.P.), or mannose and zanamivir combination, or PBS vehicle (BID, P.O.) were conducted at the indicated time post-virus-infection. Animal survival, body weight, and clinical disease were monitored for 14 days or until death. For virological and histopathological examinations, animals were sacrificed at 3−7 dpi as indicated, and their organs, tissues, and blood were collected for analyses. The two halves of harvested tissues were used for viral titration by RT−qPCR or a plaque assay and for histopathological analysis following fixation in 10% PBS-buffered formaldehyde, as we previously described[37]. For histopathological assessment, a semiquantitative system was employed to evaluate the relative degree of inflammation and tissue damage[37].

### Measurement of vital signs in animals

Real-time vital sign monitoring, including blood oxygen saturation (% of functional arterial hemoglobin), breath rate (breaths per minute, bpm), pulse distention (micrometer, μm), and heart rate (beats per minute, bpm) were measured by pulse oximetry using the MouseOx Plus (Starr Life Sciences Corp, USA).

### Metabolic assays

Seahorse XF Analyzers measure oxygen consumption rate (OCR) and extracellular acidification rate (ECAR) of live cells in a 96-well plate, interrogating key cellular functions such as mitochondrial respiration and glycolysis. The experiments were done utilizing the XF96 Seahorse

Biosciences Extracellular Flux Analyzer as previously described[38]. For seahorse glycolysis stress test (ECAR), human primary cells, hNEc and SAEc, were seeded in XF-96 cell culture plates at $1 \times 10^4$ cells/well, while A549-WT or A549-$PMI^{/-}$ were seeded in XF-96 cell culture plates at $3 \times 10^4$ cells/well, treated with 0, 5 mM, or 25 mM mannose overnight, and followed by H1N1 infection (2MOI). At 12hpi, the media was removed and replaced with unbuffered "Seahorse medium" (XF Assay Base Medium with 2 mM L-glutamine) at pH 7.4. Afterwards, the cells were treated with 10 mM glucose, 1 μM oligomycin, and 50 mM 2-DG at the indicated time points. The data were recorded and normalized by cell number. For Agilent Seahorse XF Cell Mito Stress Test (OCR), cells were treated with a similar condition as for the ECAR assay. After 24-h mannose treatment, cells were replaced with assay medium containing 1 mM pyruvate, 2 mM glutamine, and 10 mM glucose. 1.5 μM oligomycin, 1 μM FCCP, and 0.1 μM Rot/AA were injected to the cells, respectively, at the indicated time points. The data were recorded and normalized by cell number (Hoechst 33342 staining). The membrane potential of mitochondria was analyzed with the potential dependent fluorescent dye MitoProbe™ JC-1 assay kit (Thermo, M34152). ROS in the mouse lung was measured using CellROX Deep Red reagents according to the protocol of manufactures (Life Technologies, Cat#C10422).

## Metabolomics and transcriptomics studies

**Metabolomics study.** MS-based targeted analysis of fatty acids and polar metabolites was performed in the Proteomics and Metabolomics Core (LKS Faculty of Medicine, The University of Hong Kong). In terms of targeted quantitation of fatty acids, the samples were extracted by a modified Folch extraction procedure. Specifically, 4.9 ml of a Chloroform/Methanol solution (2:1, v/v) was added to the cells. The mixture was subjected to two cycles of sonication at 10 microns for 20 s on ice, followed by a 10-second pause. Subsequently, the mixture was centrifuged at $2000 \times g$ for 5 min, resulting in the formation of a clear supernatant. To this, 2.5 ml of the supernatant was combined with 0.5 ml of NaCl/Water (0.73%, w/v) and vortexed for 30 s. The resulting aqueous layer was discarded, and the organic layer was washed twice with 0.5 ml of Methanol/Water (1:1, v/v) without mixing. The aqueous layer was again discarded, and the organic layer was dried under a gentle stream of nitrogen at 45 °C. For esterification, the dried sample was dissolved in a mixture of 0.1 ml of chloroform, 1 ml of methanol, and 50 μl of concentrated hydrochloric acid (35%, w/w). The solution was then covered with nitrogen and the tube was tightly closed. After vortexing, the tube was heated at 100 °C for 1 h. Once the solution cooled to room temperature, 1 ml of hexane and 1 ml of water were added for extraction of fatty acid methyl esters (FAMEs). The tube was vortexed, and after phase separation, up to 1 μl of the hexane phase was injected for GC-MS analysis. GC/MS chromatogram was acquired in SCAN and SIM mode in an Agilent 7890B GC - Agilent 7010 Triple Quadrupole Mass Spectrometer system as previously described[39]. The sample was separated through an Agilent DB-23 capillary column (60 m × 0.25 mm ID, 0.15 μm film thickness) under constant pressure at 33.4 psi. The GC oven program started at 50 °C (hold time 1 min) and was increased to 175 °C at a ramp rate of 25 °C/min. The temperature was then raised to 190 °C (hold time 5 min) at a ramp rate of 3.5 °C/min. Finally, the temperature was raised to 220 °C (hold time 11 min) at a ramp rate of 2 °C/min. Inlet temperature and transfer line temperature were 250 °C and 280 °C respectively. Characteristic fragment ions (m/z 55, 67, 69, 74, 79, 81, 83, 87, 91, 93, 95, 96, 97, 115, 127, 143) were monitored in SIM mode throughout the run. Mass spectra from m/z 50-350 were acquired in SCAN mode.

In terms of Targeted quantitation of polar metabolites[40], the dried residue samples were redissolved and derivatized for 2 h at 37 °C in 40 μl of methoxylamine hydrochloride (30 mg/ml in pyridine) followed by trimethylsilylation for 1 h at 37 °C in 70 μl MSTFA with 1% TMCS. The qualified samples undergo the GC-MS/MS chromatogram acquiring in SCAN and MRM mode in an Agilent 7890B GC - Agilent 7010 Triple Quadrapole Mass Spectrometer system (Santa Clara, CA, USA). The sample was separated through an Agilent (Santa Clara, CA, USA) DB-5MS capillary column (30 m × 0.25 mm ID, 0.25 μm film thickness) under constant flow at 1 mL per min. The GC oven program started at 60 °C (hold 1 min) and was increased 10 °C per min to 120 °C, then 3 °C per min to 150 °C, and then 10 °C per min to 200 °C and finally 30 °C per min to 280 °C (hold 5 min). Inlet temperature and transfer line temperature were 250 °C and 280 °C respectively. Characteristic quantifier and qualifier transitions were monitored in MRM mode during the run. Mass spectra from m/z 50-500 were acquired in SCAN mode. Data analysis was performed using the Agilent MassHunter Workstation Quantitative Analysis Software.

**$^{13}$C Metabolic flux analysis (MFA).** The day after seeding, cells were washed three times with abundant PBS before adding glucose-free DMEM (supplemented with 10% FBS, 2 mM glutamine, 100 units per ml of penicillin and 100 μg per ml of streptomycin). Depending on the experiments, this medium could contain 5 mM of 1,2-$^{13}C_2$-glucose alone, together with 0, 5 mM, or 25 mM $^{13}C_6$-mannose. Cells were plated in the $^{13}$C labeled medium overnight and then infected with H1N1 for the indicated times in control or mannose-containing medium. After 12 hours-post infection (hpi), cells were harvested for $^{13}$C Metabolic flux analysis (MFA). To each sample, add 1 mL of extraction reagent consisting of methanol, acetonitrile, and water in a ratio of 4:4:2. Put the sample in an ice water bath and perform 5 cycles of ultrasonic treatment for 1 min followed by a 1-min pause. Let the sample stand at −40 °C for 30 min, then centrifuge it at $1500 \times g$ for 15 min at 4 °C. Take 1 mL of the supernatant extract and dry it using mild nitrogen, then dissolve the dried material in 50 μl of 50% acetonitrile aqueous solution (1:1, volume ratio) for UHPLC-HRMS testing. Metabolic Flux Analysis was conducted by Ultra High Performance Liquid Chromatography (UHPLC) – High Resolution Tandem Mass Spectrometry (HRMS/MS), ThermoFisher QE HF-X platform. Throughout the analysis process, the samples were placed in an autosampler at 8 °C. The samples were separated using an ACQUITY UPLC BEH Amide (2.1 × 100 mm, 1.7 μm) chromatographic column in a sample ultra-high-performance liquid chromatography system. The injection volume was 2 μL, column temperature was set at 30 °C, and the flow rate was maintained at 0.4 mL/min. The chromatographic gradient elution program was as follows: Positive ion mode: 0–0.5 min, B: 95%; 0.5–7 min, B: 95–65%; 7–8 min, B: 65–50%; 8–9 min, B: 50%; 9–9.1 min, B: 50–95%; 9.1–12 min, B: 95%. Negative ion mode: 0–0.5 min, B: 95%; 0.5–7 min, B: 95–65%; 7–8 min, B: 65-50%; 8–9 min, B: 50%; 9–9.1 min, B: 50–95%; 9.1–12 min, B: 95%. Each sample was analyzed in positive ion mode using electrospray ionization (ESI). After UHPLC separation, the samples were subjected to mass spectrometry analysis using a Thermo QE HF-X mass spectrometer. The mass spectrometry conditions were as follows: ionization source: ESI ion source; sheath gas flow rate: 30; auxiliary gas: 10; spray voltage: 2.5KV ( + )/2.5KV (-); S-Lens RF: 50; capillary temperature: 325 °C; auxiliary gas temperature: 300 °C; collision energy (NCE): 30; isolation window 1.5 m/z, Top $N = 8$. The scanning range was set from 70 to 1050 m/z, and the scanning mode was negative ion scanning. Data preprocessing was acquired with Thermo Xcalibur software (version 4.0.27.19). natural Isotope distribution was corrected analysis as previously reported[41].

**Transcriptomic study.** The lung epithelial A549 cells were pre-treated with 25 mM mannose or PBS control overnight. The cells were then infected with H1N1 (2MOI) and incubated in DMEM medium with 25 mM mannose or PBS control. At 24hpi, total cellular RNAs of the virus-infected-PBS, virus-infected-mannose, or non-infected groups ($n = 3$) were collected. Sequencing libraries were constructed and sequenced in the Proteomics and Metabolomics Core (LKS Faculty of Medicine, The University of Hong Kong). After filtering the low-quality reads, clean reads were mapped to reference sequences using HISAT/Bowtie2. The DEGs in treated samples were submitted to DAVID server for pathway enrichment and cluster analysis. Gene Ontology (GO) enrichment

analysis of DEGs was implemented using the GOseq R package. KOBAS software was used to examine the statistical enrichment of DEGS in The Kyoto Encyclopedia of Genes and Genomes (KEGG) pathways.

## Determination of antiviral EC$_{50}$ and cytotoxicity CC$_{50}$

Plaque reduction assay was performed to plot the 50% antiviral effective dose (EC$_{50}$) as we previously described with slight modifications[3]. Briefly, indicated cells were seeded at $4 \times 10^5$ cells/well in 12-well tissue culture plates on the day before carrying out the assay. After 24 h of incubation, 50–100 plaque-forming units (PFU) of the virus were added to the cell monolayer with or without the addition of drug compounds and the plates were further incubated for 1 h at 37 °C in 5% CO$_2$ before removal of unbound viral particles by aspiration of the media and washing once with DMEM. Monolayers were then overlaid with media containing 1% low melting agarose (Cambrex Corporation, New Jersey, USA) in DMEM and appropriate concentrations of individual compound, inverted and incubated as above for another 72 h. The wells were then fixed with 10% formaldehyde (BDH, Merck, Darmstadt, Germany) overnight. After removal of the agarose plugs, the monolayers were stained with 0.7% crystal violet (BDH, Merck) and the plaques were counted. The percentage of plaque inhibition relative to the control (i.e. without the addition of compound) wells were determined for each drug compound concentration. The CellTiterGlo® luminescent assay (Promega Corporation, Madison, WI, USA) was performed to detect the cytotoxicity of the selected drug compounds as previously described[34]. Briefly, cells as indicated ($4 \times 10^4$ cells/well) were incubated with different concentrations of the individual compound for 48 h, followed by the addition of substrate and measurement of luminescence 10 min later. The EC$_{50}$ and CC$_{50}$ of the drug compounds were calculated by Sigma plot (SPSS) in an Excel add-in ED50V10.

## Pseudotype-based entry inhibition assay

Vesicular Stomatitis Virus (VSV) pseudotyped with spike proteins of SARS-CoV-2 was generated as previously reported[34]. To determine the effect of PMI on viral entry, indicated cell lines, including HEK293-ACE2 cells, VeroE6-TMPRSS2, A549 cells transiently transfected with human ACE2, were infected with 100 TCID$_{50}$ respective pseudotyped VSV in the presence of treated drugs or control. The activity of firefly luciferase was measured using luciferase assay (Promega, CAT#E1501) for quantitative determination at 24 h post-transduction. Cell viability was measured via CellTiter-Glo® Luminescent Cell Viability Assay (Promega, CAT#G7571) as normalization.

## Statistical analysis

Data were analyzed using GraphPad Prism 7 (GraphPad Software, San Diego, CA, USA). Statistical significance was determined using a one-way ANOVA with Dunnett's post hoc test for multiple comparisons or Student's $t$ tests for two groups. $p < 0.05$ were considered statistically significant.

## Reporting summary

Further information on research design is available in the Nature Portfolio Reporting Summary linked to this article.

## Data availability

The Omics data described in this manuscript have been deposited in MetaboLights under accession code MTBLS9368, GEO database under accession code GSE255604, and Mendeley Data including RNA-Seq analysis (https://doi.org/10.17632/gptx4bd7ss.1); metabolomics performed in hBTEc model (https://doi.org/10.17632/z6fjtbrtn5.1); metabolomics performed in human lung A549 cell model (https://doi.org/10.17632/7ky724g4jf.1); and $^{13}$C metabolic flux analysis performed in A549 cells (https://doi.org/10.17632/5z6cjhtfth.1). Other supporting raw data are available from the corresponding author upon reasonable request. Source data are provided with this paper.

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

## Acknowledgements

This study was partly supported by the Health and Medical Research Fund (COVID1903010–Project 15, 20190732, and 21200562 to S.Y.) and the HMRF Fellowship Scheme (07210107 to S.Y.), the Food and Health Bureau, The Government of the Hong Kong Special Administrative Region; the National Natural Science Foundation of China/Research Grants Council Joint Research Scheme (N_HKU767/22 to S.Y.); the National Natural Science Foundation of China (32322087 and 32300134 to S.Y. and General Program 82272337 to J.F.-W.C.); Guangdong Natural Science Foundation (2023A1515012907 to S.Y.); Health@InnoHK, Innovation and Technology Commission, the Government of the Hong Kong Special Administrative Region (to K.-Y.Y.); the Collaborative Research Fund (C7060-21G to J.F.-W.C.) and Theme-Based Research Scheme (T11-709/21-N to D.-Y.J.) of the Research Grants Council, The Government of the Hong Kong Special Administrative Region; Programme of Enhancing Laboratory Surveillance and Investigation of Emerging Infectious Diseases and Antimicrobial Resistance for the Department of Health of the Hong Kong Special Administrative Region Government (to J.F.-W.C. and K.-Y.Y.); Emergency COVID-19 Project, Major Projects on Public Security, National Key Research and Development Program (2021YFC0866100) (to D.-Y.J., J.-F.W.C., and K.-Y.Y.); Sanming Project of Medicine in Shenzhen, China (SZSM201911014) (to K.-Y.Y.); the High Level-Hospital Program, Health Commission of Guangdong Province, China (to J.F.-W.C.); the research project of Hainan Academician Innovation Platform (YSPTZX202004) (to F.Y., J.F.-.W.C., and K.-Y.Y.); Emergency Collaborative Project of Guangzhou Laboratory (EKPG22-01) (to D.-Y.J., J.F.-W.C., and K.-Y.Y.); and the National Key R&D Program of China (projects 2021YFC0866100 and 2023YFC3041600 to D.-Y.J., J.F.-W.C., and K.-Y.Y.); and donations from the Shaw Foundation Hong Kong, Richard Yu and Carol Yu, Michael Seak-Kan Tong, May Tam Mak Mei Yin, Lee Wan Keung Charity Foundation Limited, Providence Foundation Limited (in memory of the late Lui Hac Minh), Hong Kong Sanatorium & Hospital, Hui Ming, Hui Hoy and Chow Sin Lan Charity Fund Limited, the Chen Wai Wai Vivien Foundation Limited, Chan Yin Chuen Memorial Charitable Foundation, Marina Man-Wai Lee, the Hong Kong Hainan Commercial Association South China Microbiology Research Fund, Perfect Shape Medical Limited, Kai Chong Tong, Tse Kam Ming Laurence, Foo Oi Foundation Limited, Betty Hing-Chu Lee, Ping Cham So, and Lo Ying Shek Chi Wai Foundation. The funding sources had no role in the study design, data collection, analysis, interpretation, or writing of the report.

## Author contributions

S.Y. and K.-Y.Y. conceived and designed the study. R.L., Z.Q., Z.-W.Y., Y.X., X.Y., Q.D., H.S., P.L., K.T., B.H. and J.C. designed and/or performed experiments. H.L.-X.W., H.C., J.S., H.G.-S.L., F.Y., D.-Y.J. and J.F.-W.C. provided academic advice and/or critical reagents. F.Y., D.-Y.J., J.F.-W.C., K.-Y.Y. and S.Y. acquired funding. S.Y. and K.-Y.Y. supervised the study. S.Y. and K.-Y.Y. wrote the manuscript, and all authors reviewed and edited the paper.

## Competing interests

The authors declare no competing interests.

## Additional information

[1]Academician Workstation of Hainan Province, Hainan Medical University-The University of Hong Kong Joint Laboratory of Tropical Infectious Diseases, Key Laboratory of Tropical Translational Medicine of Ministry of Education, Haikou, Hainan, China. [2]State Key Laboratory of Emerging Infectious Diseases, Carol Yu Centre for Infection, Department of Microbiology, School of Clinical Medicine, Li Ka Shing Faculty of Medicine, The University of Hong Kong, Pokfulam, Hong Kong Special Administrative Region, China. [3]School of Biomedical Sciences, Li Ka Shing Faculty of Medicine, The University of Hong Kong, Pokfulam, Hong Kong Special Administrative Region, China. [4]Department of Infectious Diseases and Microbiology, The University of Hong Kong- Shenzhen Hospital, Shenzhen, China. [5]School of Chinese Medicine, Li Ka Shing Faculty of Medicine, The University of Hong Kong, Pokfulam, Hong Kong Special Administrative Region, China. [6]Centre for Virology, Vaccinology and Therapeutics, Hong Kong Science and Technology Park, Hong Kong Special Administrative Region, China. [7]School of Chinese Medicine, Hong Kong Baptist University, Hong Kong Special Administrative Region, China. [8]Guangzhou Laboratory, Guangzhou, Guangdong Province, China. [9]Department of Microbiology, Queen Mary Hospital, Pokfulam, Hong Kong Special Administrative Region, China. [10]These authors contributed equally: Ronghui Liang, Zi-Wei Ye, Zhenzhi Qin. [11]These authors jointly supervised this work: Kwok-Yung Yuen, Shuofeng Yuan. ✉e-mail: yuansf@hku.hk

