## [Peer Review File · Nature Communications]

PMI-controlled mannose metabolism and glycosylation determines tissue tolerance and virus fitnessEditorial Note: This manuscript has been previously reviewed at another journal that is not operating a transparent peer review scheme. This document only contains reviewer comments and rebuttal letters for versions considered at Nature Communications.

Reviewers' Comments:

Reviewer #1:

Remarks to the Author:

The authors have satisfactorily addressed my concerns when this manuscript was evaluated for Nature and revised the manuscript accordingly. Therefore, this manuscript meets the high quality standards of Nature Communication.

Reviewer #2:

Remarks to the Author:

There are still unaddressed issues with the metabolic experiments that impact the conclusions presented in the manuscript. Please see my comments below.

Comments

1. The authors have not addressed the technical and interpretive issues with the metabolomics data. For example, they have still not performed any tracing experiments, yet they suggest mannose inhibits glucose fueled glycolysis. The authors need to use labeled glucose and mannose to determine whether the flux through glucose to glycolytic and TCA cycle intermediates is inhibited in both the A549 and hBTEC cells. Their metabolite pools data is also inconsistent with mannose inhibiting glycolysis – metabolites upstream of pyruvate should be decreased if this was the case. They also do not explain how their data is normalized – is it cell number, protein content, cell volume? The impact of the study is therefore uncertain.
2. The authors state that the metabolomics data are consistent between the A549 cells and the hBTECS, yet it is not possible to validate that claim as the hBTEC data is only shown using a pathway analysis. Log2FC of metabolites with statistics should be used for both.
3. The complete abolition of OCR and ECAR in A549 cells infected with H1N1 virus is still a concern that is not addressed by normalizing to cell number. The OCR and ECAR experiments should be conducted in multiple cell lines to ensure that any results are not simply cell line specific. Proliferation curves should also be shown for multiple days to ensure the cells are growing. This is also important to know when interpreting the metabolomics experiments.
4. The mitochondrial membrane potential flow cytometry data is still difficult to interpret. The CCCP control in Figure 3F is now gated differently to the other conditions. Gating the experimental conditions to this could alter the conclusions.
5. The authors now use ANOVA in their statistical tests, but it is still unclear how they performed these analyses. Were post hoc tests used?

Reviewer #3:

Remarks to the Author:

The authors have done a very good job responding to the comments of the reviewers. The manuscript describes a potentially useful therapeutic intervention in respiratory virus infections and also a novel role for PMI in viral replication. However, this reviewer still thinks that the manuscript would be clearer if it was split into two manuscripts. PMI is critical for the effects of mannose but in the absence of mannose has a proviral effect. The authors distinguish the two aspects of PMI biology, but they are not well linked.

Other comments.

1. Line 231, Figure 3A. The effects of mannose on B cell populations are shown. However, the figure itself requires further clarification. The markers used for delineation of B cells are not described and the abscissa and ordinate scale are not clear. The results may represent preferential proliferation or survival of B cells. This should be clarified.

2. Line 276, Figure 4D. The orange bar in the left hand panel should be mannose+ F6P+, based on the flow plots.

-PMI over expression diminished the effects of mannose supplementation. The authors should provide a brief explanation for this result since PMI mediates mannose function. Is this postulated to be because of its proviral effect? If so, this should be stated.

Point-to-point response

Reviewer #1 (Remarks to the Author):

The authors have satisfactorily addressed my concerns when this manuscript was evaluated for Nature and revised the manuscript accordingly. Therefore, this manuscript meets the high quality standards of Nature Communication.

Response: Thank you!

Reviewer #2 (Remarks to the Author):

1. There are still unaddressed issues with the metabolic experiments that impact the conclusions presented in the manuscript. Please see my comments below.

1.1 The authors have not addressed the technical and interpretive issues with the metabolomics data. For example, they have still not performed any tracing experiments, yet they suggest mannose inhibits glucose fueled glycolysis.

Response: We appreciate this reviewer's advice and have now used ^{13}C metabolic flux analysis (new Figures 2e and 2f) to demonstrate that the ratio of glucose-derived glycolytic metabolites was decreased upon mannose treatment. In this tracing experiment, 1,2- $^{13}\text{C}_2$ -glucose and $^{13}\text{C}_6$ -mannose labeled cells were analyzed.

1.2 The authors need to use labeled glucose and mannose to determine whether the flux through glucose to glycolytic and TCA cycle intermediates is inhibited in both the A549 and hBTEC cells.

Response: Thanks for the comments. Due to the difficulty in culturing up large amount of hBTEC primary cells (e.g. 10^7 cells/sample * 4 groups * 5 technical repeats/group = at least 10^8 cells per independent experiment), we used A549 cells in the tracing experiment (i.e., ^{13}C -labeled metabolic flux analysis, MFA). We found that glucose-derived glycolytic intermediates decrease after mannose treatment (same as our Response 1.1), whereas glucose-derived TCA intermediates increases (new Figure 2f). We then performed ECAR and OCR assays in two primary cell types (human Small Airway Epithelial Cells and human Nasal Epithelial Cells) to measure the real-time

rates of glycolysis and mitochondrial TCA cycle. Consistently, both two primary cell models exhibit decreased ECAR and increased OCR, which corroborates the MFA results in A549 cells (new Figures 2g, 2h, 3a and 3b as shown below). In view of the decreased TCA metabolites by metabolomics analysis (Figure 2d) versus a reversed pattern after MFA analysis (Figure 2f), it indicates that mannose may not only compete with glucose metabolism, but also affect utilization of other carbon sources (e.g. glutamine, amino acids, urine etc.) to maintain mitochondrial health upon virus infection.

1.3 Their metabolite pools data is also inconsistent with mannose inhibiting glycolysis – metabolites upstream of pyruvate should be decreased if this was the case.

Response: Thanks for raising up this question. Though the statistical analysis of some metabolites were less significant, general trend of reduction of the detectable glycolytic metabolites were consistent, and in a dose-dependent mannose (0 vs 5mM vs 25mM mannose). In particular, the glucose-derived G6P and lactic acid were significantly

suppressed by mannose (***) ($p < 0.001$, Figure 2f). Potential explanation to answer this reviewer might be i) cross-talk from other metabolic pathways which are mannose-mediated but non-glucose-derived can contribute to individual metabolite distribution as shown in Figure 2d; and/or ii) mannose may modulate glycolytic enzymes' activities upstream of pyruvate production.

1.4 They also do not explain how their data is normalized – is it cell number, protein content, cell volume? The impact of the study is therefore uncertain.

Response: Thanks for the comment. We have now clarified in the manuscript that the metabolomics data was normalized by cell number before sample lysis (Figure 2d Figure legend). The ECAR and OCR results were normalized by Hoechst 33342 nucleic acid staining after mannose treatment (Figure 2g Figure legend).

2. The authors state that the metabolomics data are consistent between the A549 cells and the hBTECS, yet it is not possible to validate that claim as the hBTEC data is only shown using a pathway analysis. Log₂FC of metabolites with statistics should be used for both.

Response: Thank you for raising up this issue. We have now provided the Log₂FC data with p value in Supplementary Figure 1.

3. The complete abolition of OCR and ECAR in A549 cells infected with H1N1 virus is still a concern that is not addressed by normalizing to cell number. The OCR and ECAR experiments should be conducted in multiple cell lines to ensure that any results are not simply cell line specific.

Response: We agree with the reviewer and have performed additional OCR and ECAR experiments using two primary cells, i.e., human Small Airway Epithelial Cells (SAEc) and human Nasal Epithelial Cells (hNEc). Both primary cells are physiological relevant and permissive to influenza A H1N1 virus infection. We have also repeated the same experiments in A549 cell model. We found that the ECAR were consistently down-regulated in two primary cells (new Figures 2g and 2h) and A549 cells (new

Supplementary Figure 3a and 3b), whereas OCR was up-regulated in two primary cells (new Figures 3a and 3b) but down-regulated in A549 cells (new Supplementary Figure 3c and 3d). The discrepancy in OCR between different cell types, which was correctly predicted by this reviewer, may be caused by different virus replication kinetics in cancer cell vs primary cells, as well as the different timing of irreversible and virus-induced mitochondrial damage occurs in various cells.

4. Proliferation curves should also be shown for multiple days to ensure the cells are growing. This is also important to know when interpreting the metabolomics experiments.

Response: Thanks for the constructive comments and we have performed additional proliferation curves using human primary hNEc and SAEc (new Supplementary Figure 5 and as shown below). Two conditions, i.e., with or without 10% FBS, were examined for A549 cells; whereas only one condition, i.e., with 10% FBS addition, was applied to primary cells because it is essential to maintain primary cell viability. In general, primary cells can still proliferate in two different models, though slightly less robust comparing mannose treated (25mM) and non-treated groups (0mM mannose). Moreover, comparable good cell viability can still be maintained for at least 4 days between 25mM vs 0mM mannose groups. Note the end-point of all the metabolic assays are within 24h after virus infection, thus we believe the conclusions are certain.

5. The mitochondrial membrane potential flow cytometry data is still difficult to interpret. The CCCP control in Figure 3F is now gated differently to the other conditions. Gating the experimental conditions to this could alter the conclusions.

Response: Thanks for pointing out this mistake. We have now applied the same gating strategy across different groups. The conclusion remains the same. Please refer to new Figure 3h for details.

5. The authors now use ANOVA in their statistical tests, but it is still unclear how they performed these analyses. Were post hoc tests used?

Response: Yes, we used the one-way ANOVA followed by Dunnett's multiple comparisons test. We have now clarified this in the figure legends.

Reviewer #3 (Remarks to the Author):

The authors have done a very good job responding to the comments of the reviewers. The manuscript describes a potentially useful therapeutic intervention in respiratory virus infections and also a novel role for PMI in viral replication. However, this reviewer still thinks that the manuscript would be clearer if it was split into two manuscripts. PMI is critical for the effects of mannose but in the absence of mannose has a proviral effect. The authors distinguish the two aspects of PMI biology, but they are not well linked.

Response: Thank you for the positive feedback on our revised manuscript. As correctly pointed out by this reviewer, PMI has two sides depending on whether exogenous supplement of mannose or not. Because the manuscript is based on host-virus interaction, we wish to provide a holistic picture of PMI biology by demonstrating its roles affecting both host (i.e., mannose metabolism) and virus (virus surface protein glycosylation). To strengthen the virus aspect, we performed intervention experiments to validate the proviral role of PMI as shown in Figure 5. Nevertheless, we are open to retain or remove the antiviral part after hearing from the Editor's preference.

1. Line 231, Figure 3A. The effects of mannose on B cell populations are shown. However, the figure itself requires further clarification. The markers used for delineation of B cells are not described and the abscissa and ordinate scale are not clear. The results may represent preferential proliferation or survival of B cells. This should be clarified.

Response: Thank you for pointing out this issue. As shown below, the Y axis represents the intensity of ZA staining (Zombie Aqua viability kit) to label the live cells and the X axis represents the intensity of CD1d-FITC. The gating strategy represents the effects of Mannose on CD1d positive B cells population, which was significantly increased after 24 hours of treatment. We have also performed two more independent experiments to confirm the results. Please refer to the new Figure 3c for details.

2. Line 276, Figure 4D. The orange bar in the left hand panel should be mannose+ F6P+, based on the flow plots.

-PMI over expression diminished the effects of mannose supplementation. The authors should provide a brief explanation for this result since PMI mediates mannose function. Is this postulated to be because of its proviral effect? If so, this should be stated.

Response: Thanks for pointing out this error. We have modified the figure accordingly. In terms of why PMI over expression diminished the effects of mannose supplementation, there are two possibilities. One is that as this reviewer postulated, more PMI would fuel more virus production via glycosylation thus antagonize the beneficial protection of mitochondrial health rendered by mannose; the other one is that PMI facilitates the transfer of p-mannose to p-fructose, which rescued the glycolysis suppressed by mannose. We have added this explanation in the revised context line 285-287.

Reviewers' Comments:

Reviewer #2:

Remarks to the Author:

The authors have attempted to address my comments, and although there are still concerns about their mechanism of mannose causing reduced glucose usage, the amount of work and potential impact of the data warrants publication